# MyoD-Induced Trans-Differentiation: A Paradigm for Dissecting the Molecular Mechanisms of Cell Commitment, Differentiation and Reprogramming

**DOI:** 10.3390/cells11213435

**Published:** 2022-10-31

**Authors:** Cecilia Battistelli, Sabrina Garbo, Rossella Maione

**Affiliations:** Department of Molecular Medicine, Sapienza University of Rome, Viale Regina Elena 324, 00161 Rome, Italy

**Keywords:** MyoD, myogenic conversion, forced differentiation, chromatin regulation

## Abstract

The discovery of the skeletal muscle-specific transcription factor MyoD represents a milestone in the field of transcriptional regulation during differentiation and cell-fate reprogramming. MyoD was the first tissue-specific factor found capable of converting non-muscle somatic cells into skeletal muscle cells. A unique feature of MyoD, with respect to other lineage-specific factors able to drive trans-differentiation processes, is its ability to dramatically change the cell fate even when expressed alone. The present review will outline the molecular strategies by which MyoD reprograms the transcriptional regulation of the cell of origin during the myogenic conversion, focusing on the activation and coordination of a complex network of co-factors and epigenetic mechanisms. Some molecular roadblocks, found to restrain MyoD-dependent trans-differentiation, and the possible ways for overcoming these barriers, will also be discussed. Indeed, they are of critical importance not only to expand our knowledge of basic muscle biology but also to improve the generation skeletal muscle cells for translational research.

## 1. Introduction

Trans-differentiation, also referred to as direct somatic lineage conversion [1,2], is a process in which a differentiated cell type is induced to change identity towards another differentiated state, without passing through a progenitor stage [3]. Trans-differentiation studies have several scopes, ranging from the dissection of the molecular pathways of commitment and differentiation for basic cell biology knowledge, to the generation of large amounts of a desired differentiated cell type (starting from patient derived cells) for genetic disease modeling, drug discovery and therapeutic purposes in regenerative medicine. Decades of research and countless studies involving diverse original cell types and diverse target cell fates have shown that trans-differentiation can be accomplished by means of different approaches [2,4,5]. The most direct, efficient and widely used strategy for inducing a specific cell lineage gene expression pattern consists in the ectopic expression of transcription factors specific to that lineage, using several delivery methods. As an alternative to transgene expression, some studies demonstrated the possibility of activating the regulatory regions of the endogenous factors through CRISPR/Cas9-based transcriptional regulation [6,7]. Trans-differentiation can be also attained through the manipulation of signaling or metabolic pathways, which ultimately affect the transcription factors’ activity and, even more efficiently, through the exposure to small signaling molecules combined with the introduction of lineage-specific factors [8]. More recently, increasing attention is being devoted also to microRNAs and to their regulatory networks as potential drivers of trans-differentiation [9].

A different route in the change of cell identity is the reprogramming of somatic cells into induced pluripotent stem cells (iPSC) that have the potential to differentiate, at least in principle, in almost any cell type [10], as reported in many excellent reviews on this topic (see for example [11,12,13,14,15]).

The present review will be focused on the molecular mechanisms underlying the direct route of skeletal muscle conversion of somatic cells driven by the skeletal muscle differentiation factor MyoD. The extensive knowledge learned not only on the molecular mechanisms of myogenesis but also on the general principles underlying commitment, differentiation and dedifferentiation processes will be highlighted.

MyoD holds a special place in the history of cell reprogramming, being the first transcription factor found capable of directly converting differentiated cell types into a different lineage, the skeletal muscle [16]. MyoD was discovered many years ago as a result of a series of elegant experiments initiated with the analysis of myogenic differentiation in heterokaryons [17,18,19] and with the observation that mouse fibroblast cells treated with the DNA demethylating agent 5-azacytidine were stably converted to chondrogenic, adipogenic and myogenic cells [20,21,22]. These findings, interpreted with the possible derepression of regulatory loci coding for lineage-specific transcription factors, were followed by the screening of cDNA libraries from 5-azacytidine-derived myoblasts, which led to the cloning of the myogenic determination gene number 1 (MyoD) [23]. It was subsequently found that ectopic MyoD expression could induce skeletal muscle differentiation markers not only in fibroblasts, which are of mesodermal derivation, like myoblasts, but even in some cell types derived from the two other germ layers [24,25].

At that time, the value of the MyoD discovery was to provide not only an additional and direct confirmation of the plasticity of the differentiated state, but also the first evidence that a single transcription factor can set up a complex program of gene expression leading to the acquisition of a differentiated phenotype. On the basis of this insight, a great effort was, and still is, dedicated to the identification of other transcription factors capable of reprogramming the cell fate. In this regard, a number of studies reported the successful induction of neuronal cell types [26,27] and of cardiac myocytes [28,29], starting from more or less distantly related cell types. Other works reported lineage conversions between blood cells [30,31] and between endodermal lineages [32]. Some common features shared by trans-differentiation factors emerged from these studies. For example, many of them are lineage-specific transcription factors required for critical developmental phases. More importantly, the majority of them are “pioneer” factors; indeed, they are capable of accessing closed chromatin, a crucial property that accounts for their ability to overcome the restrictions present in the differentiated cell to be reprogrammed [33]. Furthermore, they act as master regulators that induce cascades of key downstream factors cooperating in lineage specification and differentiation [34,35]. Remarkably, however, while most of the reprogramming processes require the combined expression of several transcription factors, MyoD is sufficient, even when expressed alone, to induce the myogenic conversion of somatic cells. Why MyoD is more powerful with respect to other reprogramming factors in inducing trans-differentiation is still an open question. Some possible explanations are to be sought in the MyoD’s capability of engaging a complex network of co-factors, to activate positive feedbacks and to exploit diverse epigenetic strategies that cooperate for dramatically changing the transcriptional program of the cells to be converted.

## 2. Transcriptional Activation by MyoD

MyoD is a muscle-specific member of the basic helix–loop–helix (bHLH) class of proteins, a large family of transcription factors recognizing short DNA sequences (CANNTG), termed E-box motifs, in the regulatory regions of target genes [36]. The MyoD protein (UniProt codes P10085 and P15172 for mouse and human proteins, respectively) contains a basic domain, involved in DNA binding, which is a HLH domain involved in dimerization with other HLH proteins, and two less conserved N-terminal and C-terminal domains, involved in transcriptional activation [37,38]. MyoD binding to E-boxes and the subsequent transactivation require heterodimerization with ubiquitous bHLH E-proteins, such as E12 and E47 [39]. The specificity of DNA binding and target activation by MyoD is determined by the cooperation of several mechanisms, such as the preference for internal and flanking sequences of E-boxes, the heterodimerization partners, the cooperation with other transcriptional regulators and the pre-existing chromatin accessibility [40,41,42,43]. Some transcription factors, such as MEF2 family members [44,45], Sp1 [46,47], Pbx [48] and Six proteins [49], cooperate with MyoD by directly binding to adjacent sites. A widespread role of these factors in MyoD co-regulation was further supported by studies based on the genome search for sequence motifs associated with MyoD binding regions [50,51]. A number of other factors, identified as MyoD partners by proteomic approaches [52,53] could be indirectly recruited by the myogenic factor to its target sites and modulate its activity. MyoD also interacts with the basal transcription machinery [54,55] and with the transcription elongation factor b, P-TEFb [56,57]. However, as it will be explained in detail below, a crucial role in MyoD co-activation is played by the physical and functional interaction of the myogenic factor with the epigenetic machinery.

MyoD binding and activity can be also negatively regulated. The Id proteins, which contain the HLH but not the basic domain, antagonize MyoD binding by sequestering E proteins, thus preventing the formation of active heterodimers [36,58]. Other MyoD inhibitors, such as MyoR/musculin [59,60], Twist proteins [61,62] and Mist1 [63], are bHLH proteins. These factors affect MyoD function through a combination of mechanisms, including the sequestration of E proteins, the competition with MyoD for DNA binding and the heterodimerization with MyoD into non-functional complexes. Other examples are represented by the zinc finger proteins Snail [64] and ZEB1 [65], which inhibit MyoD function by binding to the same E-boxes, where they recruit co-repressors that reduce chromatin accessibility.

Importantly, MyoD-dependent transcription is linked with extracellular cues through several signaling cascades. The best characterized is the p38 MAP kinase pathway, which promotes MyoD activity by targeting co-factors such as MEF2 and E proteins, as well as MyoD-interacting chromatin complexes [66,67,68]. On the other hand, the activation of the Notch pathway inhibits MyoD function through the Notch effector Hes1/Hey1, a bHLH transcriptional repressor that competes with MyoD for E-box binding [69]. Several mitogenic signals also inhibit MyoD function through multiple mechanisms, among which are the up-regulation of Id proteins and the activation of cyclin/cdk complexes that directly target MyoD [70]. This is why efficient in vitro differentiation of myogenic cells requires the withdrawal of serum from the culture medium. Interestingly, while the presence of mitogens during myogenic conversion prevents terminal differentiation, it does not seem to prevent the commitment step to the myoblast precursor; as revealed by a genome-wide transcriptome analysis, in this condition, the up-regulation of myoblast-specific and the down-regulation of cell-of-origin networks are observed [71]. This finding should be addressed more in depth, as it would mean the possibility of separately studying the commitment and differentiation steps.

MyoD-induced trans-differentiation involves the activation of a complex program of gene expression, which starts with the activation of direct targets such as the bHLH muscle-specific transcription factor myogenin and the co-activator MEF2 [72]. MyoD also induces its own transcription [73] and the expression of other transcription factors [72]. These factors, in turn, activate downstream muscle-specific genes, acting through amplifying cascades or through feed-forward mechanisms involving the cooperation with MyoD itself [72,74,75]. Trans-differentiation by MyoD is associated not only with the acquisition of the muscle phenotype by the cell undergoing conversion, but also with the loss of the original phenotype [24,25]. Indeed, global analyses of gene expression during fibroblast-to=muscle conversion revealed that many, though not all, fibroblast-related genes are down-regulated [71,76]. Although MyoD is a transcriptional activator, it can repress some genes negatively regulating myogenesis through the induction of microRNAs (e.g., MyoD induces the expression of miR-206, which targets Fstl1 and Utrn, suppressed during skeletal muscle differentiation) [77]. Moreover, MyoD is capable of keeping some muscle-specific genes in proliferating myoblasts repressed until the occurrence of a differentiation signal, through the recruitment of chromatin-compacting enzymes [78] (see also below). However, the molecular mechanisms by which MyoD can turn off the fibroblast differentiation program and, even harder to explain, any original differentiation program are still puzzling.

A schematic representation of the best characterized co-factors and signaling pathways that promote or inhibit the transcriptional activation by MyoD is illustrated in Figure 1.

The discovery of MyoD was followed by the identification of three other bHLH muscle regulatory factors (MRFs) structurally and functionally related to MyoD and exclusively expressed in skeletal muscle: myogenin [79], Myf5 [80], and MRF4 [81]. In vivo studies addressing the spatio-temporal expression pattern of MRFs and the effects of their genetic ablation on muscle development, differentiation and regeneration, highlighted both overlapping and specific roles for these factors. The general picture emerging was that MyoD and Myf5 are involved in the commitment of muscle precursors to the myogenic lineage, myogenin in terminal differentiation and MRF4 in both phases of myogenesis [82,83,84]. Although all of them were shown to induce skeletal muscle markers when ectopically expressed in fibroblast cells, none of them were able to drive myogenic conversion as efficiently as MyoD [85,86]. Comparative analyses for genome-wide binding, induction of chromatin modifications and regulation of gene expression have been focused on MyoD, Myf5 and myogenin [87,88], but not yet on MRF4. These studies suggested that Myf5 is more active as a chromatin modifier than as a transcriptional activator, myogenin is principally a transcriptional activator, while MyoD possesses both a potent transactivation domain and the ability to initiate chromatin modifications.

## 3. Chromatin Regulation by MyoD

The chromatin status, extremely important for transcriptional control during the changes of cell identity, is determined by several regulatory layers, such as the density and positioning of nucleosomes, the different histone post-translational modifications and the folding of the chromatin fiber into higher-order structures of different length scales. The dynamics of these regulatory layers, driven by the interplay between cis-regulatory elements, transcription factors, chromatin-modifying enzymes and long noncoding RNAs, are strictly interdependent on each other [89,90,91].

One of the key properties by which MyoD can initiate cell reprogramming is believed to be its ability to contact E-boxes within closed chromatin, the typical feature of the so-called “pioneer” transcription factors [33]. Actually, MyoD was supposed to have a weak pioneer activity due to the structural features of its bHLH domain [92]. In fact, it has been found that, during differentiation, the binding of MyoD to the myogenin promoter is preceded by the binding of the homeodomain protein Pbx [48]. Pbx would allow the recruitment of the myogenic factor to a non-canonical E-Box by physically interacting with MyoD; then, chromatin modifications, increased accessibility of the adjacent canonical E-boxes, stable MyoD binding and further chromatin remodeling at the regulatory region allow the induction of myogenin expression [48,93,94]. However, Pbx is required for several but not all MyoD-regulated genes [42,95]. Moreover, a genome-wide analysis of chromatin accessibility in embryonic stem cells through ATAC-seq and MNase-seq showed that MyoD can bind to sites previously embedded in closed chromatin and subsequently induce chromatin modifications [96].

Regardless of the assistance of other factors in accessing chromatin, MyoD is capable of dramatically reorganizing the chromatin landscape of trans-differentiating cells by engaging and coordinating multiple epigenetic mechanisms (detailed below and schematized in Table 1 and Figure 2).

**Table 1 cells-11-03435-t001:** Key factors exploited by MyoD for reprogramming the chromatin state.

Class	Recruited or Targeted Factor	MyoD-Induced Effect	References
Nucleosome remodeling factors	SWI/SNF complex	Relaxation of nucleosome positioning at MyoD targets	[93,97,98]
	CHD2	Incorporation of the histone variant H3.3 and marking of muscle promoters for activation	[99]
Histone acetylases and deacetylases	p300/CBP	Histone acetylation and transcriptional activation	[72,100,101]
	pCAF	MyoD acetylation and transcriptional activation	[102,103]
	HDAC I	Histone deacetylation and inhibition of premature activation of MyoD targets	[78]
Histone methylases and demethylases	Set7/9	Accumulation of H3K4me1 and assembly of active muscle enhancers	[50,104]
	Prmt5	H3R8 dimethylation and increased recruitment of the SWI/SNF complex	[105]
	LSD1	Demethylation of H3K9me2 and derepression of MyoD targets	[106]
	Utx	Demethylation of H3K27me3 and derepression of MyoD targets	[107]
Long noncoding RNAs	SRA	Cooperation with MyoD-induced gene expression	[108,109]
	Linc-RAM	Support to the assembly of the MyoD-SWI/SNF complex on the regulatory regions of muscle genes	[110]
	LncMyoD	Increase of chromatin accessibility at MyoD binding sites	[111]
	Kcnq1ot1	Displacement of EZH2 and release of gene repression at the p57kip2 locus.	[112]
Architectural proteins mediating chromatin folding	CTCF	Regulation of long-distance chromatin contacts mediated by CTCF	[71,113,114,115]

### 3.1. Interaction with Nucleosome Remodeling Complexes

The stability and position of nucleosomes can be altered by chromatin remodeling enzymes that, by assembling, disassembling or sliding nucleosomes, can cover or uncover the binding sites for transcription factors [116]. Early evidence that MyoD acts by modifying chromatin accessibility for promoting transcription came from trans-differentiation experiments showing the relaxation of nucleosome positioning at the promoters of MyoD targets [86]. It was subsequently discovered that a key role in this process is played by the ATP-dependent chromatin-remodeling complex SWI/SNF, which is bound and recruited by MyoD on muscle gene promoters [93,97]. SWI/SNF complexes contain either the Brg1 or the Brm1 ATPase subunit and a variable number of different BAF (Brg1/Brm-associated factors) structural subunits. SWI/SNF complexes are typically involved in sliding and ejecting nucleosomes, thus exposing binding sites for transcriptional activators or repressors. The molecular mechanism by which MyoD directs the SWI/SNF complex to its targets involves the preassembly of a complex containing MyoD and the muscle-specific BAF60c subunit on gene promoters before differentiation, followed by the recruitment of a Brg1-containing SWI/SNF complex upon differentiation stimuli causing BAF60c phosphorylation by p38 kinase [98].

An additional mechanism of MyoD-induced chromatin remodeling involves the direct interaction of the myogenic factor with the CHD2, a member of the CHD subfamily of chromatin remodelers involved in nucleosome editing [99]. By recruiting CHD2 prior to differentiation, MyoD induces the incorporation of the histone variant H3.3 genome wide at myogenic gene promoters, marking them for activation [99].

As also highlighted below, the recruitment of chromatin remodeling complexes is further reinforced by the MyoD-induced chromatin modifications, providing one of the numerous examples of the positive feedbacks activated by the myogenic factor.

### 3.2. Interaction with Histone-Modifying Enzymes

A wide range of histone modifications including acetylation, methylation, phosphorylation, and ADP-ribosylation affect the degree of local chromatin condensation [89]. Increased knowledge about the epigenetic enzymes involved in the deposition of these modifications and improved methods for mapping their genome-wide distribution partially clarified their relationship with the chromatin structure at promoters and enhancers and with the transcriptional dynamics. Most information concerns the acetylation and methylation of specific residues of H3 and H4 histone tails. For example, acetylation of lysine 9 in histone H3 (H3K9Ac), acetylation of lysine 20 in histone H4 (H4K20Ac), acetylation of lysine 27 in histone H3 (H3K27Ac), mono- or dimethylation of lysine 4 in histone H3 (H3K4me1/2) and tri-methylation of lysine 4 in histone H3 (H3K4me3) are generally associated with chromatin relaxation; in contrast, tri-methylation of lysine 9 in histone H3 (H3K9me3), tri-methylation of lysine 27 in histone H3 (H3K27me3) and tri-methylation of lysine 20 in histone H4 (H4K20me3) are associated with chromatin compaction.

One of the first evidences that MyoD cooperates with histone-modifying enzymes was the interaction of the myogenic factor with the histone acetyltransferase (HAT) p300/CBP [100,101] and the increase of histone acetylation at some MyoD targets in concomitance with MyoD binding and gene activation [72]. Histone hyperacetylation, in turn, reinforces the chromatin accessibility of MyoD targets by promoting the recruitment of the chromatin remodeling SWI/SNF complex [93]. MyoD also associates with the HAT pCAF, but this interaction is mainly involved in its acetylation, a post-translational modification required for MyoD function [102,103,117]. Interestingly, MyoD was found to recruit class I histone deacetylases (HDAC I) to the myogenin promoter and to repress its activation in proliferating, undifferentiated myoblasts [78]. This mechanism would serve for preventing premature differentiation until the replacement of HDACs with HATs. The possible role of HDACs in MyoD-dependent commitment during trans-differentiation, however, is not known. Large scale approaches demonstrated that MyoD binds genome-wide to promoters, enhancers and thousands of additional intergenic sites in undifferentiated myoblasts, differentiated myotubes, as well as in trans-differentiated fibroblasts, and that its binding is associated with the hyperacetylation of histones H3 and H4 [51]. However, the observation that MyoD binding and histone modifications did not necessarily correlate with transcriptional activation suggested a more complex role for MyoD in the reorganization of chromatin architecture (see below).

Another chromatin-modifying enzyme that interacts with MyoD is the histone methyltransferase Set7/9 [104], which promotes the accumulation of H3K4me1, a typical chromatin signature of enhancer regions [118]. Set7/9 is recruited by MyoD on muscle regulatory regions and is required for efficient trans-differentiation of fibroblasts to myoblasts [104]. In this regard, the integration of MyoD-binding profile data with the genome-wide distribution of specific chromatin signatures, such as H3K4me1 and H3K27Ac, highlighted the critical role of MyoD in the assembly of active enhancers for the activation of muscle-specific genes [50].

A further histone-modifying enzyme used by MyoD for inducing permissive chromatin at its targets is Prmt5, a protein arginine methyltransferase [105]. Through direct interaction with MyoD, Prmt5 binds to the myogenin promoter, in which it introduces the dimethylation of histone H3 arginine 8 (H3R8). This modification promotes the recruitment of the Brg1 subunit of the SWI/SNF chromatin-remodeling complex and is required for chromatin remodeling and transcriptional activation [105]. Whether the interaction between MyoD and Prmt5 is involved in the wider effects of MyoD binding to the genome has not been investigated yet.

Along with the deposition of activating histone modifications, the induction of MyoD targets also involves the removal repressive chromatin marks such as the methylation of H3K9 and the methylation of H3K27. This is in part due to the decreased expression and/or to signal-dependent post-translational modifications of the enzymes responsible for these modifications. Down-regulation of repressing chromatin enzymes has been reported for: Suv39H1, which catalyzes H3K9me3 [119]; G9a, which catalyzes H3K9 mono and dimethylation; as well as MyoD methylation [120] and EZH2, the catalytic subunit of the polycomb repressive complex 2 (PRC2), which catalyzes H3K27me3 [121,122]. In addition, the removal of repressive methylation from MyoD targets also involves the ability of the myogenic factor to recruit histone demethylases such as LSD1 [106], which targets H3K9me2, and UTX [107], a H3K27me3-specific demethylase. Interestingly, it has been reported that treatment with 2-hydroxyglutarate, which prevents H3K9 demethylation at MyoD targets, inhibits MyoD-mediated trans-differentiation of mouse fibroblasts but not the differentiation of already committed C2C12 myoblasts. Although the putative MyoD-interacting demethylase whose inhibition causes this effect is not known, this finding suggests its specific involvement in the commitment step of the differentiation process [123].

### 3.3. Interaction with Long Noncoding RNAs

A large number of long noncoding RNAs, both ubiquitous and muscle-specific, have been involved in either promoting or restricting skeletal muscle differentiation. These long noncoding RNAs are localized in the nucleus and/or in the cytoplasm and affect myogenesis through diverse mechanisms resulting in chromatin modifications, transcriptional modulation or post-transcriptional regulation [124,125,126]. Some of the long noncoding RNAs demonstrated as important in myogenesis have been shown to be direct MyoD partners for achieving the transactivation of its targets.

The first long noncoding RNA found to be involved in muscle differentiation was steroid receptor RNA activator (SRA), which physically interacts with MyoD and with the associated RNA helicases p68/p72 at muscle promoters [108]. Depletion of SRA impairs MyoD-induced gene expression and myogenic conversion, indicating a critical role of the observed interaction for MyoD activity [108,109]. The molecular mechanism underlying the cooperation between SRA and MyoD has not been defined. However, it is worth mentioning that SRA can interact with trithorax group (TrxG) or polycomb repressive complex 2 (PRC2) complexes [127] and, interestingly, also with the cohesin complex, regulating CTCF-mediated insulation and chromatin contacts [128].

Linc-RNA activator of myogenesis (Linc-RAM) is a muscle-specific long noncoding RNA associated with chromatin and is required for the activation of a wide range of MyoD targets [110]. The molecular mechanism of this cooperation involves the physical interaction of MyoD with Linc-RAM, which promotes the assembly of the MyoD-SWI/SNF complex on the regulatory regions of muscle genes [110].

Another long noncoding RNA, LncMyoD, is directly activated by the myogenic factor and promotes muscle differentiation through at least two distinct mechanisms [111,129]. One involves the inhibition of the IMP2-mediated translation of proliferation genes, allowing growth arrest that facilitates differentiation [129]. The other one involves the physical interaction of LncMyoD with MyoD across the genome and the increase of chromatin accessibility at the co-occupied sites [111]. Very interestingly, LncMyoD is required for the differentiation of muscle precursors and for the myogenic conversion of fibroblast cells, but not for the differentiation of already committed myoblasts, suggesting its role in the first step of MyoD-mediated trans-differentiation [111].

Further evidence that MyoD makes use of long noncoding RNAs for regulating gene expression is provided by the physical and functional interaction of the myogenic factor with the macro LncRNA Kcnq1ot1 in inducing the expression of the cdk inhibitor p57kip2 [112]. In this case, MyoD exploits the chromatin interaction with Kcnq1ot1 for displacing EZH2 and releasing gene repression.

Some other long noncoding RNAs are involved the activation of MyoD targets, but their direct interaction with MyoD has not been proved. For example, the long noncoding RNA MUNC (also called ^DRR^ RNA), a trans-acting enhancer-derived RNA transcribed from the distal regulatory region (DRR) enhancer of MYOD, cooperates with the myogenic factor by promoting MyoD binding and transactivation of some targets [130,131,132]. The long noncoding RNA Irm is also up-regulated during myogenesis and positively regulates the differentiation process. This long noncoding RNA enhances the expression of MyoD/MEF2 targets by interacting with chromatin at these loci and facilitating the assembly of MyoD/MEF2D complexes. However, Irm does not seem to interact with MyoD, but only with MEF2D [133].

With the advent of increasingly advanced methods for the computational analysis of long noncoding RNA-protein interactions and their genome-wide binding profiles, it will certainly turn out that the use of long noncoding RNAs by MyoD is a more common strategy than we thought.

### 3.4. Three-Dimensional Genome Reorganization

It has long been recognized that the spatial organization of the chromatin plays an important role in coordinating gene expression during developmental processes [134,135] and that it is regulated by a bidirectional interplay with transcription factors and chromatin modifications [136,137]. The three-dimensional (3D) organization of the genome results from hierarchical levels of chromatin folding [138,139]. The first level involves the formation of chromatin loops, whose sizes vary from kilobases to megabases, and which are involved in several functions including promoter-enhancer interactions. A higher order level involves the folding into topologically associated domains (TADs), which are characterized by the presence of boundaries that constrain the spreading of repressive chromatin and behave as co-regulatory units that limit the genomic interactions within the domain. Both loops and TADS can be stabilized by CCCTC-binding factor (CTCF) in cooperation with the cohesin complex [139]. At a further level, multiple TADs can then associate with each other to form the so-called A and B compartments, euchromatic and heterochromatic, respectively.

The study of MyoD-dependent commitment and differentiation has now provided important insights into the molecular mechanisms underlying the bidirectional interplay between transcription factors and chromatin folding. Early evidence that MyoD can regulate gene expression by altering the spatial organization of chromatin was provided by the regulation of individual MyoD targets. For example, it was reported that MyoD induces the expression of the cdk inhibitor p57kip2 by disrupting a CTCF-mediated chromatin contact of p57 promoter with a repressive regulatory element located about 150 kilobases far from the gene [140]. The molecular mechanism involves the physical and functional interaction of MyoD with CTCF, which results in the displacement of cohesin complex subunits [113]. Moreover, MyoD was shown to drive the formation of repressive inter-chromosomal interactions between the regulatory regions of some muscle genes expressed at late times of myogenesis [114]. These interactions, which are induced by MyoD during the commitment to the myogenic lineage, keep late myogenic genes inactive until the appropriate time of differentiation, suggesting a mechanism for the temporal regulation of gene expression during the differentiation process [114].

The development of increasingly sensitive high-throughput chromatin conformation capture (Hi-C) techniques and of very high-resolution imaging methods allowed to reveal that MyoD-dependent differentiation is associated with a global reorganization of the genome architecture [71,115,141]. These studies indicated that MyoD is required for the establishment of genome-wide chromatin contacts between regulatory elements by binding to promoters, enhancers and insulators during the commitment step of fibroblast trans-differentiation [71], as well as in differentiating myoblasts [115]. It is worth mentioning that MyoD binding resulted in transcriptional repression and altered long-range interactions between the regulatory elements of TGF-β1, a fibroblast-related gene [71]. This suggests that the reorganization of the three-dimensional genome architecture may be important in the strategy by which MyoD represses the gene expression program of the cell of origin during reprogramming.

The molecular mechanisms by which transcription factors drive chromatin topology are just beginning to be clarified [136,137] and include, among others, the contribution of the ongoing transcription and/or of the local chromatin structure. However, MyoD-dependent chromatin looping precedes the activation of gene expression and is not altered by the inhibition of transcription [71] nor by the modulation of MyoD-induced H3K27ac levels [115]. The integration of the Hi-C maps with the ChIPseq data for MyoD and CTCF led to the suggestion that MyoD alters the chromatin contacts by functionally interacting with CTCF at loop anchor regions [71,115]. Interestingly, MyoD was also found to function in vitro as an anchor protein, by itself, and suggested to mediate muscle-specific chromatin loops even independently of CTCF [115], but this property has not been further explored.

## 4. Limits to MyoD-Dependent Trans-Differentiation

Since its discovery, it was realized that MyoD, despite its reprogramming potency, was unable to induce the myogenic conversion of some cell types such as embryonal carcinoma cells, HeLa cells and some hepatocyte-derived cell lines [24,142]. Many efforts have also been made for converting embryonic stem cells (ES cells) or induced pluripotent stem cells (iPS cells) through exogenous MyoD expression [143,144]. These approaches, aimed at possible applications of patient-derived cells in regenerative medicine, revealed the existence of several constraints to the MyoD-mediated conversion of mouse ES cells [145], human ES cells [146] and human iPS cells [147,148]. The observed limitations, which are the object of extensive investigation, reflect the existence of epigenetic barriers, the lack of trans-acting co-factors (or the presence of trans-acting repressors) and the occurrence of abnormal signaling. For example, the unresponsiveness of HeLa cells to MyoD-mediated conversion was ascribed to the deficiency of Baf60c and of p38 kinase activity [98]. Importantly, the manipulation of this and other epigenetic pathways significantly improved the scarce efficiency of myogenic conversion of human ES cells and iPS cells. In this regard, it was reported that the introduction of exogenous Baf60c allows MyoD to promote chromatin remodeling at myogenic targets and to induce muscle gene expression in embryonic stem cells [146]. Moreover, the exogenous expression of the histone demethylase JMJD3 (KDM6B), which removes di- and tri-methylation from H3K27, combined with that of MyoD, triggers the myogenic differentiation of human-induced pluripotent stem cells, although H3K27 demethylation seems to indirectly affect the activation of muscle-specific genes [147].

It is worth considering that trans-differentiation is not complete even when MyoD is ectopically expressed in fibroblasts, the cell type most amenable to myogenic conversion [76,149].

Epigenetic constraints were described for specific MyoD targets, such as cdkn1c, which is not induced in some fibroblast cell types, while it is in others [150]. The analysis of this model system revealed that an interplay between pre-existing DNA methylation and H3 K9 dimethylation prevents MyoD binding and the consequent chromatin reorganization required for gene activation [151,152].

Another mechanism limiting MyoD function involves poly(ADP-ribose) polymerase 1 (PARP1). PARP1 was reported to keep some paradigmatic MyoD target genes under control during the differentiation process by interfering, independently of the enzyme activity, with MyoD binding and with the accumulation of the activating histone modification H3K4me3 [153]. This kind of interference was observed in differentiating myoblasts, but it is conceivable that PARP1-dependent chromatin features may play a more general role in restricting MyoD function during trans-differentiation processes.

Incomplete trans-differentiation of fibroblast cells was explored at a genome-wide scale trough an integrated analysis of gene expression, MyoD binding and chromatin accessibility in MyoD-converted fibroblasts compared to myoblasts [76]. This work revealed that the failure to up-regulate a number of muscle-specific and to down-regulate a number of fibroblast-specific genes was associated with deficiencies in chromatin remodeling. However, the exact relationship between pre-existing chromatin features, MyoD binding and chromatin accessibility was not completely characterized in this study. The recent employment of single-cell transcriptome analysis associated with pseudo-temporal ordering of cells not only confirmed the incomplete reprogramming of human fibroblasts but also allowed the reconstruction of differentiation trajectories during MyoD-mediated conversion [149]. The authors developed a computational method for comparing the pseudo-temporal trajectories of MyoD-converted fibroblasts with those of normal human myoblasts and revealed some of the barriers that divert fibroblast cells from the muscle fate to alternative paths. This approach revealed that MyoD fibroblasts fail in the up-regulation of an IGF1-mediated autocrine signaling, known to support MyoD function through multiple mechanisms, and over-secrete BMP proteins, known to impair MyoD activity by autocrinally inducing ID [149].

Very interestingly, the expression of ectopic MyoD in combination with a cocktail of three small molecules (the cyclic AMP agonist forskolin, the TGF-β inhibitor RepSox and the GSK3-β inhibitor CHIR99021) was shown to reprogram mouse fibroblast cells into induced myogenic progenitor cells (iMPCs), highly similar to satellite cells but more easily expandable and capable of spontaneously differentiating into contracting myotubes [154]. A detailed analysis of the transcriptomic and epigenomic features of the direct conversion to muscle by MyoD alone, compared with the indirect conversion through iMPCS by MyoD plus small molecules, revealed that the two cell fates are reached through distinct molecular trajectories and that the transition through iMPCS leads to a more stable and complete differentiation with respect to the direct route [155,156]. There is no doubt that these studies highlighted a novel approach for producing reliable muscle stem cells for several types of translational applications. However, the molecular mechanism by which MyoD, when combined with small molecules, causes dedifferentiation is not yet clear. In particular, the transcriptional and epigenetic co-factors used by MyoD for the activation of stem cell markers, as well as for the repression of fibroblast cell markers, deserve further investigations.

## 5. Concluding Remarks

As highlighted throughout the review, one of the most significant outcomes of dissecting the strategies by which MyoD alone succeeds in reprogramming the cell identity was the disclosure of a number of molecular mechanisms underlying the plasticity of the differentiated state and the transcriptional regulation of differentiation processes. These studies led to the identification and characterization of networks involving transcriptional regulators, chromatin interactions, long noncoding RNAs and signaling pathways, in some cases specific to the muscle system but in other cases common to other lineages.

As introduced above, one of the scopes of forcing differentiation towards a specific lineage is the production of functional differentiated cells for therapeutic applications and disease modeling. After an initial excitation regarding the possibility of using ex vivo MyoD transfer in regenerative medicine, it became clear that the direct lineage conversion does not exactly recapitulate the program of muscle development, especially as regards the down-regulation of the resident gene expression program and the physiological maturation of myofibers. Furthermore, importantly, cells expressing exogenous MyoD tend to be directly driven to terminal differentiation, thus losing the regeneration potential and the property to be expanded. In this regard, intensive research is currently focused on developing protocols, based on the expression of myogenic regulators combined with epigenetic modulators and signaling molecules, for generating muscle progenitors and for improving their differentiation and regeneration potential [143,144]. Further efforts are being addressed toward skeletal-muscle tissue engineering through emerging biotechnologies including self-organized 3D skeletal muscle organoids and scaffold-based platforms mimicking muscle architecture [157].

Nevertheless, it is worth pointing out that exogenous MyoD expression in both pluripotent and somatic cells provides several advantages and opportunities. Exogenous, regulated MyoD expression allows the generation of a large amount of terminally differentiated cells in a simple, fast and inexpensive manner. In this regard, MyoD-induced differentiation of patient-derived iPS cells proved to be useful for recapitulating some of the phenotypes of Duchenne muscular dystrophy and other muscle disorders [148,157]. A deeper exploration of MyoD-induced dedifferentiation of easily accessible somatic cells toward the myoblast precursor stage followed by faithful differentiation [154,156] also appears promising, if this approach could be applicable to human cells.

In conclusion, further insights on the molecular roadblocks to MyoD-induced trans-differentiation and on the networks that facilitate the efficiency of reprogramming and of terminal differentiation are needed, not only to expand our knowledge of basic muscle biology but also to improve the current approaches for disease modeling, drug discovery and new therapeutic approaches.

## Figures and Tables

**Figure 1 cells-11-03435-f001:**
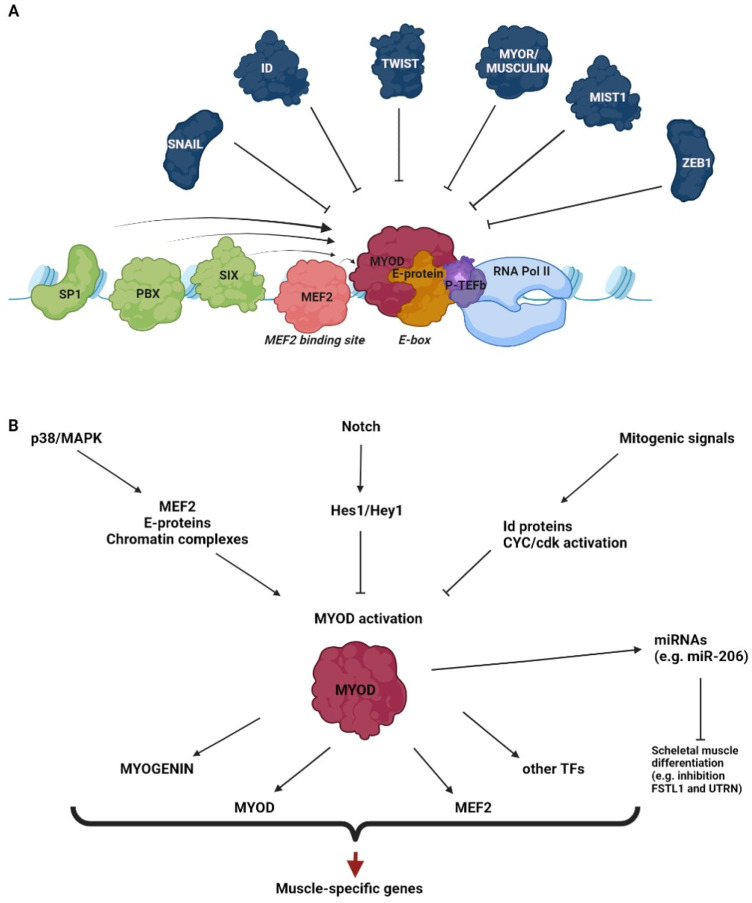
Regulation of MyoD transcriptional activity. The drawings outline some of the main co-factors (**A**) and signaling pathways (**B**) that promote or inhibit the transcriptional activation by MyoD.

**Figure 2 cells-11-03435-f002:**
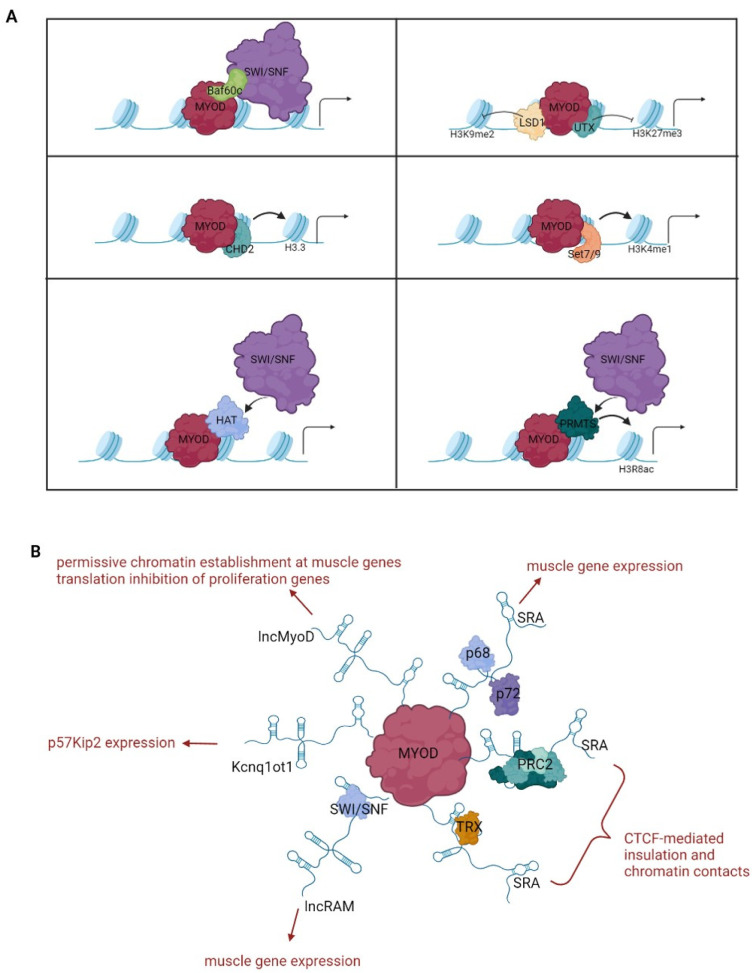
Chromatin regulation by MyoD. The drawings outline the best characterized chromatin modifying complexes (**A**) and long noncoding RNAs (**B**) exploited by MyoD for chromatin reprogramming.

## Data Availability

Not applicable.

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
