# Peer review of "MyoD-Induced Trans-Differentiation: A Paradigm for Dissecting the Molecular Mechanisms of Cell Commitment, Differentiation and Reprogramming"

_cells, 2022, doi:10.3390/cells11213435_

Round 1

Reviewer 1 Report

First, I would like to congratulate the excellent work that was presented. I understand how laborious it is to prepare a manuscript for publication and we must give proper credit. Congratulations.

You've done a good job. However, I have a few comments on this paper.

1.      As this is a review article, it would be interesting to see some figures describing those paths presented in topics 2 (Transcriptional activation by MyoD) and 3 (Chromatin regulation by MyoD). This will make the work enriched. A review that features figures related to the paths gains greater visibility and citations. Therefore, I would suggest 2-3 figures in the text.

2.      I suggest using the "BIORENDER" site to create these professional science figures.

Overall, this review article was well written. I was happy to be able to contribute some comments to this paper.

Author Response

Reviewer 1 Comments

You've done a good job. However, I have a few comments on this paper.

  1. As this is a review article, it would be interesting to see some figures describing those paths presented in topics 2 (Transcriptional activation by MyoD) and 3 (Chromatin regulation by MyoD). This will make the work enriched. A review that features figures related to the paths gains greater visibility and citations. Therefore, I would suggest 2-3 figures in the text.

  1. I suggest using the "BIORENDER" site to create these professional science figures.

Response

We thank the reviewer for the congratulations and for the useful suggestion.

We have now included two figures, created with BIORENDER. The first one schematizes the best characterized co-factors and signaling pathways that promote or inhibit the transcriptional activation by MyoD. The second one outlines the key chromatin modifying complexes and long noncoding RNAs  exploited by MyoD for chromatin reprogramming.

Reviewer 2 Report

Dear authors,

I found the paper to be overall very well written and I felt confident that you all performed careful literature analysis and research data interpretation. Minor revision of the manuscript should be considered.

Minor comments:

Line 91-94: I suggest that it would be good to include the code of the human MyoD protein from Uniprot for more information at protein level.

Line 167/168 - myf5 ? with uppercase Myf5  as in line 161;

Line 445-458- ” the expression of ectopic MyoD in combination with a cocktail of 445 three small molecules was shown to reprogram mouse fibroblast cells into induced myo-446 genic progenitor cells (iMPCs)” ..the three molecules are mentioned several times here and I consider that should to be specifically indicated.

Congratulations on a well-written review work of the current state of the art in muscle-specific transcription factor, MyoD, research.

Author Response

Reviewer 2 Comments

Minor comments:

Line 91-94: I suggest that it would be good to include the code of the human MyoD protein from Uniprot for more information at protein level.

Line 167/168 - myf5 ? with uppercase Myf5  as in line 161;

Line 445-458- ” the expression of ectopic MyoD in combination with a cocktail of 445 three small molecules was shown to reprogram mouse fibroblast cells into induced myo-446 genic progenitor cells (iMPCs)” ..the three molecules are mentioned several times here and I consider that should to be specifically indicated.

Congratulations on a well-written review work of the current state of the art in muscle-specific transcription factor, MyoD, research.

Response

We thank the reviewer for the congratulations and for the comments

We have now included the UniProt codes for both mouse and human MyoD proteins, modified myf5 as Myf5 and specified the three small molecules used in combination with MyoD for reprogramming fibroblasts to iMPCs

Reviewer 3 Report

Review deals with MyoD, a specific skeletal muscle differentiation factor which plays an important role in conversion of non-muscle somatic cells into skeletal muscle cells. Summarized is known information about the molecular mechanisms by which MyoD reprograms the transcriptional regulation of the cell of origin during the myogenic conversion.  The interaction of the myogenic factor with the epigentic machinery are also very well described. From my point of view is a very good point of authors to include into review chapter “Limits to MyoD-dependent trans-differentiation”.

Review is focused on an actual and very important topic and is well written. Presented are actual information and that in an understandable form.

Author Response

Reviewer 3 Comments

Review deals with MyoD, a specific skeletal muscle differentiation factor which plays an important role in conversion of non-muscle somatic cells into skeletal muscle cells. Summarized is known information about the molecular mechanisms by which MyoD reprograms the transcriptional regulation of the cell of origin during the myogenic conversion.  The interaction of the myogenic factor with the epigentic machinery are also very well described. From my point of view is a very good point of authors to include into review chapter “Limits to MyoD-dependent trans-differentiation”.

Review is focused on an actual and very important topic and is well written. Presented are actual information and that in an understandable form.

Response

We thank very much the reviewer for appreciating both the topic and the writing of our manuscript.
